# The Impact of Economic Crisis in Areas of Sprawl in Spanish Cities

**Arlinda García-Coll** [1,*] **and Cristina López-Villanueva** [2] 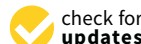

[1]   Department of Geography, University of Barcelona, 08001 Barcelona, Spain
[2]   Department of Sociology, University of Barcelona, 08034 Barcelona, Spain; clopez@ub.edu
[*]   Correspondence: arlindagarcia@ub.edu; Tel.: +34-934-037-852

**Abstract:** The development of dispersed urbanism in Spain ran parallel to the real estate boom and consolidated a new model of city sprawl based on the expansion of suburban areas. This process, which started in the mid 1980s, came to a halt with the onset of the economic crisis in 2007. With it, construction stopped, mobility fell, and urban growth came to a standstill. The purpose of this article is, firstly, to analyse the recent evolution and chronology of the expansion of dispersed urbanism in the Barcelona Metropolitan Region (BMR) in order to gain an insight into some of its explanatory factors, and secondly, to look into the future middle-term prospects of dispersed urbanism in the BMR and Spain. To this end, we examine trends in the housing market and residential mobility and take stock of the impact of business cycles on them. The conclusion is that dispersed areas still retain their appeal for people in the life stages of the creation and expansion of households. For this reason, an effective economic recovery and a renewed rise in the price of housing in denser cities may contribute to an upturn in the popularity of the dispersed residential model, which nowadays could be considered to be in a 'lethargic' phase, waiting for certain factors to concur and reactivate its expansion.

**Keywords:** dispersed urbanism; residential strategies; residential mobility; economic crisis; Barcelona Metropolitan Region; social crisis; land squandering

---

## 1. Introduction: Dispersed Urbanism in the Reconfiguration of Spanish Urban Regions; Evidence and Questions

On 23 June 2015, *The Washington Post* [1] published a news item on the evolution of European cities as derived from the analysis of their growth in population between the censuses of 2001 and 2011. The main conclusion highlighted by the article could be summarised, in the authors' own words, as "European cities are becoming more American". It is significant that an American newspaper echoed the demographic decline of European urban centres and the increase in population of their respective metropolitan areas, comparing these with American cities, where these processes have been commonplace. Even if a comparison like this should be qualified by means of theoretical tools and empirical evidence [2,3]—especially in the case of the Mediterranean countries—it is clear that the boom of dispersed urbanism in Europe has not gone unnoticed.

Català et al. [4] point out the existence of common elements in the processes of urban dispersion undergone by the European Mediterranean countries. They specifically identify the following shared features: very late occurrence of the sprawl phenomenon; extreme urban densification in previous stages; great vitality of the central cities; and a high presence of small and medium-sized cities in the country's urban system. They also point to the existence of three overlapping urban models: the traditional compact model, the dispersed model of the new residential areas, and the mixed land uses exemplified by industrial and commercial activities that tend to occupy large land containers. Salvati [5] describes this last model as "hybrid", underscoring its importance in order to understand the particularities of Mediterranean Europe.

Despite all these shared aspects, Carlutti et al. [6] have recently centred on the differences between individual Mediterranean countries to reach the conclusion that it is not possible to speak of a single model of urban dispersion in Mediterranean Europe and it is necessary to take into account both regional and local factors to understand the specific evolution of each one of the countries in the area.

Spanish urban regions are a good example of the situation described by the *Washington Post* article, as well as of the Mediterranean model of urban dispersion. On comparing the sprawling processes in twenty-six European countries in the 1990s and the first half of the 2000s, Siedentop and Fina [7] concluded that Ireland, Portugal, and Spain were the European countries which had experienced the the greatest sprawl. After several decades of urban growth based on a model of compact urban development, at the turn of the century, Spanish cities began to change their patterns of urban expansion. On the one hand, they experienced an accelerated tendency towards suburbanisation, characterised by a loss of both population and compactness in the largest municipalities and the growth of their peripheral areas, which became more extensive [8,9]. On the other hand, a new model of dispersed city was established, a model which had been rarely seen before and which was scarcely relevant among Spanish urban territorial trends, besides being mostly associated with second homes. It is this last fact that constitutes a distinctive factor of the Spanish process of urban dispersion [10]. Thus, in the 1960s, Spain saw the development of areas of residential expansion at a short distance from the cities, areas which were mostly devoted to second homes and which in more recent stages would act as the germ of subsequent dispersed urban development. According to Burriel [11], this was a phenomenon unmatched in the rest of Europe, and it turned out to be a crucial factor to be taken into account if we wish to understand the expansion of dispersed urbanism in Spain at the end of the 20th century.

Initially, and in this context of mainly vacational use and reduced population and territorial extension, the pressure that dispersed urban areas exerted on their territories—and on the services offered there—was rather low, and so the relevance of such areas was minimal. However, the situation changed from the 1980s onwards. Then, a strong growth of dispersed urbanism took place in parallel with the real estate boom, and there was a rapid increase in the number of people that chose to live permanently in such locations, that is, to live '*dispersedly*'. Proof of this transformation can be seen, for example, in the Barcelona Metropolitan Region (BMR), where it is estimated that almost one-third of the urban land subjected to development between 1993 and 2000 was allocated to dispersed residential use [12], and where the annual population growth of low-density municipalities exceeded 4% between 1999 and 2006 [13].

The increase of dispersed urbanism in Spanish urban areas was a consequence of a series of factors. In the first place, the real estate bubble was responsible for an extraordinary rise in the price of homes in the city centres, causing an increase in the demand for affordable housing with better price-to-perfomance ratios. Additionally, among the several reasons for residential mobility, there was the demand for housing in places of greater environmental quality and closer to nature, which just added up to the demand for certain specific housing conditions which were difficult to find in city centres—larger, single-family houses, with a private garden, and so forth. The phase of economic growth prior to 2008 also contributed to this whole process by bringing about lower unemployment rates, rising salaries, and above all, increased facilities to obtain a mortgage loan. All these factors have been brought up by numerous studies dealing with such residential transformations from different outlooks: the demographic perspective [12], the economic one [14,15], the social point of view [16], and the perspective of environmental impact [17].

The bursting of the real estate bubble and the beginning of the economic recession from 2007 onwards brought this phase to an end and led into a new situation characterised by the minimising of new-house building in low-density areas as well as demographic stagnation. Thus, the process of residential dispersion came to a sudden halt. This new stage has posed several short and medium-term questions concerning the role of dispersed urbanism in the socio-residential dynamics of Spanish urban areas previously affected by processes of suburbanisation. First of all, the debates have focused

on whether it is possible to speak of land squandering, or otherwise, the model of dispersed urban growth that can continue and proceed in an orderly, sustainable way. The way that the expansion of the dispersed model took place in the past—characterised by dubiously administered extremely fast growth, with an abrupt stop when the crisis set in—calls into question the viability and continuity of dispersed urbanism in cities that, apart from periods of intense property speculation, had traditionally displayed a compact type of urban growth. Moreover, the sudden decline of dispersed urban expansion brought about by the economic crisis has made students wonder whether we have just witnessed the end of a process, or rather the beginning of an impasse that will come to an end as soon as the economic situation improves. The economic crisis brought urban dispersion to a halt, but will the economic recovery trigger an upturn in the demand of housing in dispersed areas?

A second set of questions concerns the social impact of the economic crisis on the population that moved to live in dispersed areas. The higher cost of living in dispersed quarters [17], the difficulties derived from the breaking-up of neighbourhood solidarity networks as a consequence of moving to a new area [16], and the high rate of indebtedness of the families that changed their place of residence in a time of rising housing prices all make the residents in dispersed areas bound to suffer the social consequences of the crisis in a most severe way.

To answer all these questions, two different lines of enquiry are needed. First, it is necessary to undertake a revision of the past, since an analysis of past processes will certainly give us interesting clues about future possibilities. Also, it will be necessary to examine the most recent trends, especially those from 2014 on, when the Spanish government officially declared the end of the economic crisis in the country—despite the views of many microeconomics and social researchers.

The BMR is a good example of a metropolitan region where dispersed urban developement set in late and grew at an accelerated pace. Inititally, the territories were not prepared to respond to the new situations generated by the dispersed model, and thus, their adaptation to their new changing realities could only take place simultaneously to the implementation of the new urban model. By studying these features of the BMR case, the present article attempts to contribute some empirical evidence to the debates about the specificities of the establishment of dispersed urbanism outside the Anglo-Saxon regions, in line with the works of Couch et al. [18] or Richardson and Bae [2]. More particularly, it is our aim to help devise a framework which is useful in specifying the role of the economic crisis in processes of urban dispersion, according to the findings of Salvati [5] and Cho et al. [19], whose research discloses the undeniable effects of the economic crisis, which not only puts an end to construction, but also to trends of suburban development in a territory.

Also, the present article argues for the need of incorporating the perspective of demographic studies into morphological and urbanisation studies, with demographic studies focusing on specifying the elements responsible for residential dispersion, such as residential mobility or the different directions of this mobility, depending on people's stages in life. To this end, our research combines information coming from official statistics on residential mobility in Spain, with data coming from two surveys of people who moved to live in a dispersed setting. These surveys do not only inform us of the causes behind people's migration, but they also provide us with the interviewees' assessment of their change of residence and information about their future migration projects. Often, as Champion [20], Tyrell and Kraftl [21], or Coulter et al. [22] suggest, lack of data leads to analyses of the socioeconomic factors determining urban sprawl which only take into consideration the moment when people's migration take place (e.g., the analyses of Weilemmann et al. [23] for Switzerland, or Salvati [5] for Athens), without taking into account people's whole migratory course of life. This renders such analyses unable to tell us anything about future migration prospects. As Coulter et al. [22] state, the point here is to rethink residential mobility as "linking lives through time and space".

Our article deals with two main issues: On the one hand, it describes and explains the evolution of dispersed urbanism in the Barcelona Metropolitan Region (BMR) and establishes the chronology of its recent evolution based on the analysis of intrametropolitan residential mobility data (intensity, direction of flows, and characteristics of the moving persons), the observation of trends in the housing

market (new constructions) at the metropolitan level, and the impact of business cycles. Based on these analyses, it will be possible to describe in greater depth the sociodemographic challenges facing dispersed areas in the recent past and at the present moment. On the other hand, the article deals with the future middle-term prospects of dispersed urbanism in the BMR and Spain. All in all, our study seeks to improve our knowledge of the present functioning of Spanish metropolitan dynamics based on their past and most recent developments, with the ultimate goal of contributing to the administering of low-density urbanism in Spain. Nowadays, low-density urbanism occupies a prominent place in territorial planning, since the changes in land use that it entails make it a relevant factor to be considered when planning both the present of urban development and, above all, its future prospects, as it is aptly observed by Ramankutty and Coomes [24].

## 2. Materials and Methods

In this section, we shall deal with three basic issues that constitute the departing point of our research. In the first place, we will centre on methodological questions to reflect about both the theoretical and the empirical difficulties when it comes to defining dispersed urbanism. As we shall see, although it is true that dispersed urbanism has been the subject of a good number of both theoretical and applied studies, its definition and measurement have important limitations, independent of whether we consider the Spanish case or that of other countries. Secondly, we shall introduce the methodology employed in our research, which was based on a classification of municipalities specifically designed to identify those with a strong presence of dispersed urbanism. Finally, we shall proceed to the description of the sources used in our empirical analysis.

### 2.1. Methodological Approach: Defining Dispersed Urbanism

To begin with, there is no consensus definition of dispersed urbanism, which often makes it difficult to establish the necessary criteria to demarcate the spatial extent of the phenomenon. Generally, studies have used extensive land occupation [25,26]—closely related to low-density areas [27,28]—as the most common criterion for demarcating dispersed urban areas. Sometimes, besides taking into account population density, the definition of dispersed urbanism additionally involves the presence of morphologically and functionally isolated urban elements, with the prevailing type of dwelling consisting of single-family, detached, or semidetached houses [29]. After all, population density is one of the most widely used criteria for measuring residential dispersion [30]. Several recent works have used net density to study dispersed urbanism, sometimes in combination with measures of distance and/or discontinuity from the city centre based on observations using CORINE Land Cover [31]. Multidimensional measurements of urban sprawl are frequent too, generally using Geographic Information System (GIS) tools. Jaeger and Schwick [32] proposed the use of weighted urban proliferation (WUP) to consider the area built over in a given landscape (amount of built-up area), the dispersion of the built-up area, and the land uptake of the built-up area per inhabitant or job. Weillenmann et al. [23] consider that built-up areas may spill across administrative borders, with neighbouring municipalities sharing a similar employment market and mobility infrastructure and requiring coordination when it comes to spatial planning. All these sophisticated methodologies undoubtedly feed an interesting debate; however, the most important thing to do before making a decision regarding the definition of dispersed urbanism to be used in a particular study is to establish the research objectives and to evaluate if the statistical data needed are available.

In our case, the main objective revolved around demographic trends rather than spatial structures. Moreover, in the Spanish case, we came across a serious limitation concerning the potential sources of information available to researchers, namely: there exists no exact correspondence between the various administrative and statistical divisions and dispersed residential areas, especially when it comes to suburban residential complexes. Thus, it was not always feasible for us to reconstruct the sociodemographic traits of such areas simply via the aggregation of census sections or the inspection of the municipal registers of inhabitants or other sources of intramunicipality information. In actual

fact, only in the case of a few municipalities was it possible for us to gather statistical information regarding dispersed residential areas. Finally, we had to resource to fieldwork in order to complete our territorial analysis, something which proved also useful in order to put to the test the municipal typology we were using and to select the urban residential complexes where our interviews would subsequently be carried out. The combination of these three strategies produced a methodological approach properly adjusted to the objectives of our research.

*2.2. Methods: A Municipality Typology Based on Dispersed Urbanism*

In the face of all the aforementioned obstacles, our study used a methodological strategy of its own in order to define and characterise dispersed urbanism in the BMR; one which was developed in the context of the Program Research, Development and Innovation (R+ D+ i) projects entitled "Mobility, Family Solidarity, and Citizenship in the BMR" (2003–2006) and "Social Change and Urban Transformation Processes in a Context of Crisis in the Urban Peripheries of Large Metropolitan Areas in Spain. The Case of the BMR" (2014–2017). On the one hand, we used an indirect approach based on information about the municipalities taken as a whole, and on the other hand, we used two ad hoc surveys with interviews carried out in 2005 and 2017 in a sample of suburban residential areas.

First of all, to identify the phenomenon of dispersed urbanism at the municipal level, we considered the surface of urban land allocated for residential use and calculated its net density. Analysing the Swiss case, Weilenmann et al. [23] have shown that the municipality is a meaningful unit of analysis for the examination of sprawl because it is the political entity that makes decisions about urban spatial development and because it has sufficient data available. In our case, we also had information about the municipal surface allocated to "extensive, low-density patterns of land use, with single-family or two-family (semidetached) houses surrounded by a plot of land with a garden", from the Urban Planning Map of Catalonia (MUC) [33] prepared by the General Office for Country Planning and Urbanism of the Catalan Government. This allowed us to calculate the proportion of residential land surface allocated to this type of dwelling. By combining both indicators (net density and percentage of land allocated to detached/semidetached houses), we were able to group the different municipalities in the BMR in several categories according to their degree of compactness or dispersion (Figure 1). We identified five types of municipalities, categorising as 'dispersed' those situated in the first and second quintiles of the net-density distribution, i.e., with less than 140 inhabitants/hectare in 2015. The percentage of land allocated to detached/semidetached houses was used to distinguish between dispersed and highly dispersed areas as internal categories of urban dispersion (Table 1). According to 2015 data, in the BMR, there are 106 municipalities (64.7% of all municipalities) which satisfy the criteria for dispersed areas, amounting to 14.2% of the BMR population.

**Table 1.** Classification of Barcelona Metropolitan Region (BMR) municipalities by urban typology.

| Typology | Municipalities | | Population | | Net Density | Isolated Houses |
|---|---|---|---|---|---|---|
| | Number | % | Number | % | Inhab/ha (*) | % Land (**) |
| Highly Compact | 12 | 7.3 | 2,506,046 | 49.8 | >501 | <6.0 |
| Compact | 10 | 6.1 | 737,568 | 14.7 | 351–500 | 6.0–20.0 |
| Medium | 36 | 22.0 | 1,070,608 | 21.3 | 141–350 | 20.0–40.0 |
| Dispersed | 57 | 34.8 | 447,419 | 8.9 | <140 | 40.0–80.0 |
| Highly Dispersed | 49 | 29.9 | 266,617 | 5.3 | <140 | >80.0 |
| Total | 164 | 100.0 | 5,028,258 | 100.0 | 204.8 | 58.0 |
| Aggregated data | | | | | | |
| Compact | 22 | 13.4 | 3,243,614 | 64.5 | 576.7 | 10.2 |
| Dispersed | 106 | 64.6 | 714,036 | 14.2 | 50.5 | 82.3 |

(*) Computed using the surface area of urban soil allocated for residential use. (**) Percentage of land area allocated to detached/semidetached houses. Source: Compiled by the authors based on *Censo de población* (Population Census) 1991, *Padrón Municipal de habitantes* [Municipal Register of Inhabitants] 1996, and *Padrón continuo* (Continuous Register) 1998–2016, by the National Institute of Statistics (INE).

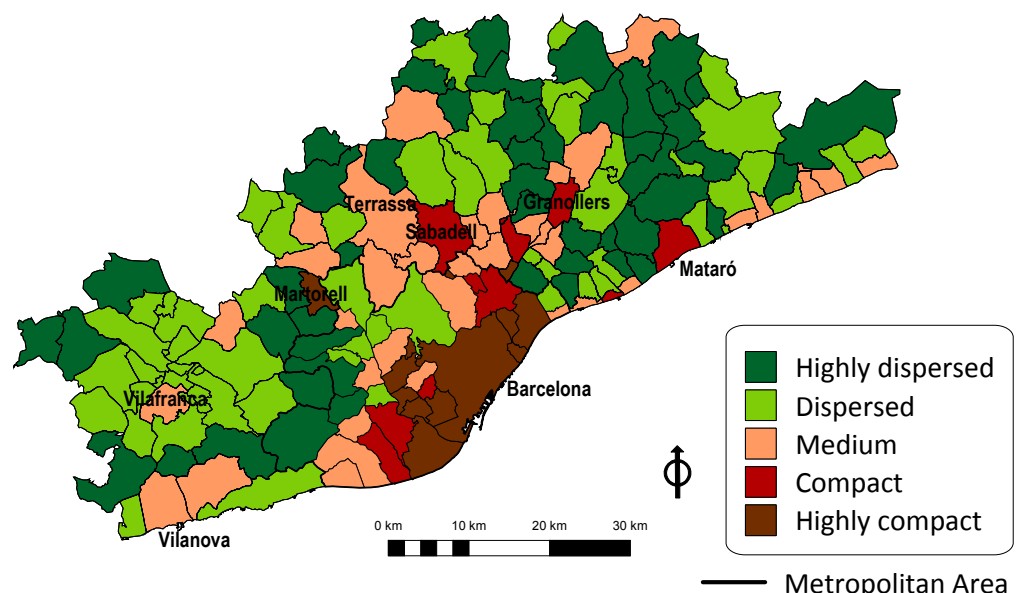

**Figure 1.** Classification of BMR municipalities by urban typology. Source: Compiled by the authors based on *Censo de población* (Population Census) 1991, *Padrón Municipal de habitantes* (Municipal Register of Inhabitants) 1996, and *Padrón continuo* (Continuous Register) 1998–2016, by the National Institute of Statistics (INE).

Once we had identified those municipalities with a strong presence of dispersed urbanism—which we termed as 'dispersed municipalities'—our analysis of their evolution was carried out based on statistical sources which provided us with individual data for each municipality. For this stage of the analysis, we grouped together the categories of 'dispersed' and 'highly dispersed' areas, since both of them satisfy the criteria established to identify municipalities with a sprawling residential pattern as an essential part of their urban morphology.

*2.3. Data Sources*

The sources used in this particular phase of the study were the microdata provided by the *Statistics of Residential Variation* (EVR) and the *Continuous Register* (Padrón continuo). The EVR allowed us to study intrametropolitan residential mobility in the period 1996–2016, and therefore, the migration flows displayed by the different types of municipality. The *Continuous Register*, on the other hand, made it possible to describe the characteristics of the populations in the different municipalities meeting the criteria for dispersed areas. Even though this was an indirect characterisation, because it included the totality of the population in the different municipalities, the trends identified provided us with a robust framework in order to delimit the existing processes.

Additionally, as a complement to this preliminary observation of our object of study, we used the results of two surveys that we ourselves carried out in 2005 and 2017. Both of them had been designed to gather information about the living standards of people residing in dispersed areas in the BMR. One section of the questionnaire was aimed at getting information about people's migratory experiences. Here, it must be taken into account that in 2005, all the interviewees had changed their municipality of residence at least once in their lifetime. More than 90% of them had lived in a compact area at some point and thus were able to compare both lifestyles. Also, all the interviewees had been living in a sprawl area for over 12 years, and so they had experienced the effects of the economic crisis while living in this type of residential area. Contrary to the basic general information provided by the Municipal Register or the EVR (which only include data about sex, age, place of birth, and nationality), our surveys allowed us to complete the sociodemographic profiles of the interviewees (profession, employment status, household income, etc.) and gave us information about their motives, assessment

of their residential area, and future migratory projects, all of which was highly valuable to achieve the goals of our research.

The first survey, called "Mobility, Family Solidarity, and Citizenship in Metropolitan Regions", included a total of 600 households (1024 individuals) from a sample of 24 suburban residential complexes in 17 municipalities. This first survey was implemented by using quotas for different socioeconomic, age, and sex categories; this way, the results were representative of the totality of the population living in suburban residential complexes in the BMR. The second survey, entitled "Social Change and Urban Transformation Processes in a Context of Crisis in the Periphery of the BMR", was a replica of the one carried out in 2005. This time, information was gathered about 1759 individuals who had been living in the selected surburban residential complexes since at least 2005. Both surveys provided us with information about the living conditions of people residing in this kind of suburban setting: their family structure, labour conditions, spaces of life, family and social relationships, reasons for moving into a suburban residential complex, and reasons for choosing the place of residence. To this, other highly significant information must be added, such as: the residents' assessment of issues related to the house they lived in and the suburban residential complex they inhabited, or to municipal policies, as well as information concerning their future residential projects. The 2017 survey was complemented with questions aimed at comparing the situations in 2005 and 2017, apart from questions concerning the interviewees' assessment of the changes in their employment status or their family income during that period, all of which was helpful for evaluating the impact on them of the economic crisis.

## 3. Results

The process of residential expansion and increase of dispersed urbanism in the Barcelona Metropolitan Region took off in the second half of the 1980s, transforming the territory in a radical way [34]. This section deals with the recent chronology of dispersed urbanism based on an analysis of the growth and evolution of its population, intrametropolitan residential mobility rates, and rates of new property construction.

*3.1. Business Cycles, Real Estate Market, and Residential Mobility: Stages in the Recent Evolution of Dispersed Urbanism in the BMR and its Explanatory Factors*

In 1991, dispersed municipalities in the BMR contained 351,340 inhabitants. In 2016, their population was 717,832 (Table 2 and Figure 2), which means that their population had doubled in that period. Besides this steep rise in population, these municipalities had drastically changed their demographic structure and composition over that same period of time.

**Table 2.** Population evolution (1991–2016) in BMR municipalities by typology.

| Typology | 1991 | 2002 | 2008 | 2016 | 1991–2001 | 2002–2007 | 2008–2016 |
|---|---|---|---|---|---|---|---|
| Highly Compact | 2,540,899 | 2,395,323 | 2,511,575 | 2,514,324 | −0.5 | 0.8 | 0.0 |
| Compact | 634,037 | 672,045 | 722,827 | 739,658 | 0.5 | 1.2 | 0.3 |
| Medium | 738,146 | 888,204 | 1,028,087 | 1,074,929 | 1.7 | 2.5 | 0.6 |
| Dispersed | 234,852 | 337,546 | 418,273 | 450,410 | 3.4 | 3.6 | 0.9 |
| Highly Dispersed | 116,488 | 189,505 | 248,090 | 267,422 | 4.5 | 4.6 | 0.9 |
| Aggregate data | | | | | | | |
| Compact | 3,174,936 | 3,067,368 | 3,234,402 | 3,253,982 | −0.3 | 0.9 | 0.1 |
| Medium | 738,146 | 888,204 | 1,028,087 | 1,072,929 | 1.7 | 2.5 | 0.6 |
| Dispersed | 351,340 | 527,051 | 666,363 | 717,832 | 3.8 | 4.0 | 0.9 |
| Total | 4,264,422 | 4,482,623 | 4,928,852 | 5,046,743 | 0.5 | 1.6 | 0.3 |

Source: Compiled by the authors based on the *Censo de población* (Population Census) 1991, *Padrón Municipal de habitantes* (Municipal Register of Inhabitants) 1996, and *Padrón continuo* (Continuous Register) 1998–2016, by Instituto Nacional de Estadística (National Institute of Statistics) (INE).

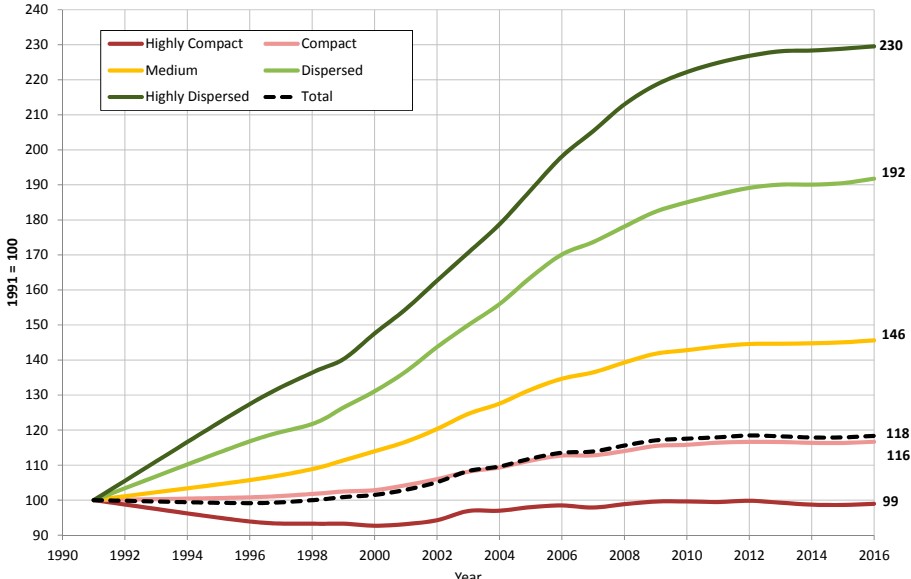

**Figure 2.** Population evolution in BMR municipalities classified by typology. Indexed annual data. Population in 1991 = 100. Source: Compiled by the authors based on the *Censo de población* (Population Census) 1991, *Padrón municipal de habitantes* (Municipal Register of Inhabitants) 1996, and *Padrón continuo* (Continuous Register) 1998–2016, by Instituto Nacional de Estadística (National Institute of Statistics) (INE).

However, this transformation process did not take place in a homogeneous way over the period, and it is possible to distinguish four stages of evolution:

(a) First, there is the stage from 1991 to 1999, when the population grew at an annual rate of slightly over 3%. At this point, we witness the consolidation of this form of residence, which had started to develop in a previous period, and the start of the boom of dispersed residential areas.

(b) A second stage comprises the period 2000–2006, when the great boom of dispersed urbanism took place in the BMR. In this phase, we see the effects of five factors. In the first place, we have a real estate market in which higher density areas were saturated, with rising prices and lack of diversity of the residential offer in the city. In that context, lower density areas offered unprecedented possibilities of population absorbtion and expansion [35]. Secondly, the type of property which was built was adressed to families with young children, fond of living in a quiet environment close to nature, with high environmental standards. Thirdly, this stage marked the beginning of an expansive business cycle which generated better economic prospects for households and created a climate of economic confidence that had an upward effect on the residential market. Fourthly, in view of the economic prosperity, banks focused their mortgage policy on the provision of credit facilities to buy a home [36]. Finally, the existence of a previous offer of suburban housing developments well-established in the territory made it easier for dispersed urbanism to expand. Although in the past, these had been of rather limited size and were mainly used for vacation purposes [35], they provided embryonic spaces for building projects.

As a result of this combination of elements, we witnessed a boom of dispersed urbanism, characterised by a very intense growth of population, with annual rates of over 4% that coincided with rates of initiation of new-property construction nearing 45% [37], and net migration rates higher than 33 per thousand persons (Figure 3 and Table 3).

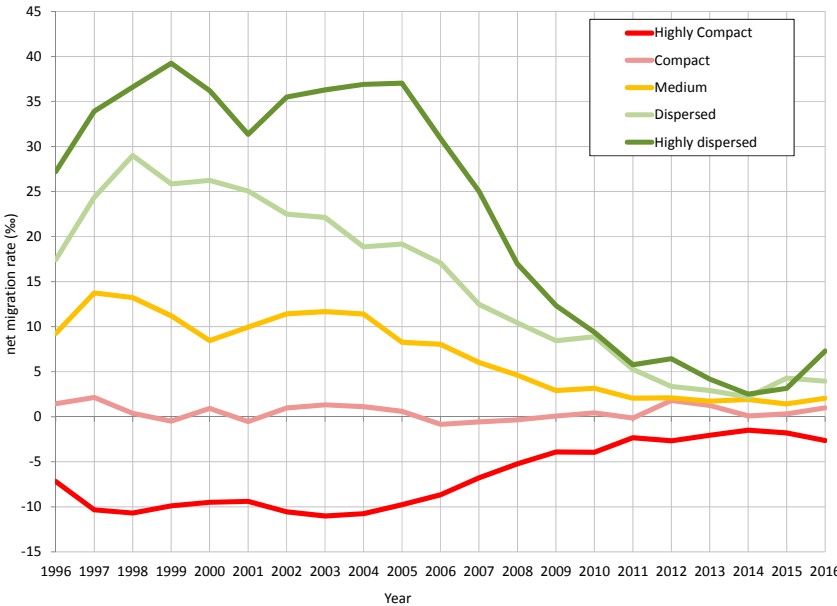

**Figure 3.** Net intrametropolitan migration rate by types of municipality in the BMR, 1996–2016. Source: Compiled by the authors based on *Padrón continuo* (Constinuous Register) 1996–2017 and *Estadística de Variaciones Residenciales* (Residential Change Statistics) 1996–2016, microdata file, INE.

(c) The third stage was marked by a profound turn away from the previous trajectory. It began with the sudden onset of the economic crisis, whose effects were gradually felt. Some authors [38,39], when dealing with the changes that took place in this period, have highlighted the advance in the metropolitanisation of the BMR.

First of all, there was a drop in the rates of new-property construction in dispersed municipalities, rates that, from 2006, went abruptly down to values of less than 5 per thousand in 2009 and thereafter.

**Table 3.** New-property construction rate by types of municipality in the BMR, 1999–2017.

| Year | New-Property Construction Rate (‰) | | | | | Single Family New-Property Construction Rate (‰) | | | | |
|---|---|---|---|---|---|---|---|---|---|---|
| | Highly Compact | Compact | Medium | Dispersed | Highly Dispersed | Highly Compact | Compact | Medium | Dispersed | Highly Dispersed |
| 1999 | 9.17 | 20.43 | 27.54 | 37.40 | 36.58 | 1.04 | 3.26 | 6.72 | 16.68 | 27.07 |
| 2000 | 8.32 | 21.32 | 29.79 | 36.51 | 38.01 | 0.49 | 1.72 | 4.56 | 11.88 | 27.78 |
| 2001 | 7.10 | 16.80 | 24.11 | 30.98 | 31.00 | 0.31 | 1.37 | 2.86 | 8.32 | 18.55 |
| 2002 | 7.55 | 17.52 | 26.03 | 29.44 | 31.79 | 0.36 | 1.11 | 2.94 | 8.10 | 17.96 |
| 2003 | 7.68 | 21.05 | 28.37 | 31.69 | 35.79 | 0.33 | 1.14 | 3.06 | 9.82 | 21.61 |
| 2004 | 8.23 | 23.15 | 32.89 | 34.58 | 47.76 | 0.18 | 0.83 | 2.96 | 9.33 | 24.74 |
| 2005 | 8.15 | 21.26 | 34.07 | 47.07 | 49.21 | 0.16 | 0.79 | 2.82 | 9.40 | 25.07 |
| 2006 | 9.73 | 27.17 | 40.75 | 44.48 | 52.62 | 0.15 | 0.86 | 3.17 | 7.48 | 23.13 |
| 2007 | 7.67 | 16.58 | 25.41 | 31.67 | 34.75 | 0.11 | 0.59 | 1.99 | 4.89 | 13.07 |
| 2008 | 4.31 | 5.57 | 6.97 | 7.74 | 10.05 | 0.07 | 0.21 | 0.69 | 1.56 | 4.77 |
| 2009 | 2.12 | 2.67 | 3.47 | 2.45 | 5.09 | 0.04 | 0.15 | 0.35 | 0.94 | 2.00 |
| 2010 | 2.85 | 4.28 | 3.29 | 6.74 | 6.63 | 0.04 | 0.21 | 0.51 | 1.20 | 2.50 |
| 2011 | 1.83 | 3.14 | 1.75 | 5.71 | 2.71 | 0.03 | 0.16 | 0.30 | 0.86 | 2.02 |
| 2012 | 1.26 | 1.46 | 1.44 | 1.53 | 1.91 | 0.03 | 0.19 | 0.28 | 0.64 | 1.47 |
| 2013 | 0.84 | 0.33 | 0.70 | 1.36 | 1.80 | 0.03 | 0.08 | 0.27 | 0.54 | 0.99 |
| 2014 | 1.19 | 1.60 | 0.92 | 0.97 | 1.08 | 0.03 | 0.11 | 0.33 | 0.61 | 0.91 |
| 2015 | 1.95 | 1.60 | 1.71 | 3.42 | 2.00 | 0.04 | 0.27 | 0.37 | 0.88 | 1.40 |
| 2016 | 2.56 | 2.31 | 2.44 | 5.05 | 2.65 | 0.05 | 0.27 | 0.82 | 1.27 | 2.09 |
| 2017 | 3.40 | 4.55 | 4.45 | 7.17 | 3.22 | 0.05 | 0.45 | 0.72 | 1.75 | 2.36 |

Source: Compiled by the authors based on Generalitat de Catalunya Departament de Territori i Sostenibilitat (Department of Territory and Sustainability) *Licencias viviendas iniciadas y acabadas* (Building Licenses for Started and Finished Houses) and *INE Censos de viviendas* (Housing census), 2001 and 2011.

Secondly, in-migration came to a sudden standstill (Figure 4), with the subsequent effects on net migration rates, which fell to values of under 10 per thousand from 2009 onwards. A certain degree of saturation in the housing supply accounts for the fact that the first sector to experience the economic

downturn was the construction sector, whose downfall preceded the drop in sales. The rise in the value of land once it was put to use increased the price of housing, and this slowed down the pace of new construction. Next, the effects of the economic crisis pulled down the rates of migration of people motivated by the search for a better dwelling due to an increase in the requirements to obtain a mortgage [36] and the readjustment of family budgets in the context of rising unemployment and wage settlements in the case of employed individuals [40]. Overall, property sales became stagnant due to the fall in arrivals of new residents, and this dragged the construction of new houses down to a minimum level.

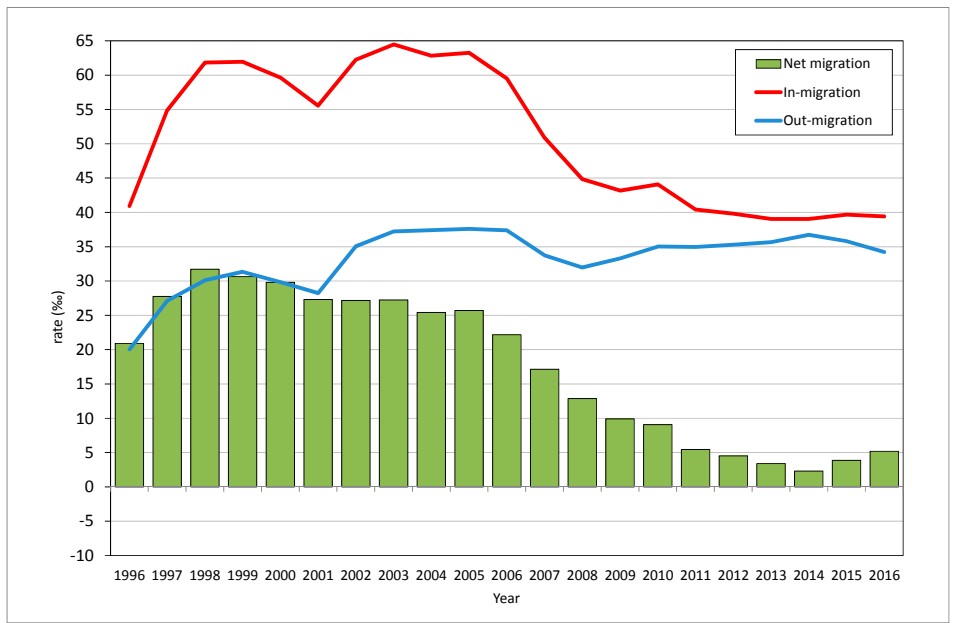

**Figure 4.** Component elements of the intrametropolitan migration dynamics of dispersed municipalities (1996–2016). Source: Compiled by the authors based on the *Padrón continuo* (Continuous Register) 1996–2017 and the *Estadística de Variaciones Residenciales* (Residential Change Statistics) 1996–2016, microdata file, INE.

Despite the gradual shrinking of the real estate market and the drop in the arrival of in-migrants, there was a certain inertia in the evolution of the growth of dispersed municipalities, which remained high until 2012 (Figure 4). This dynamic was a consequence of natural population growth in the context of a high birth rate due to the inflow of young adults. For this reason, it was not until 2012 that a phase of demographic stagnation set in.

(d) The fourth stage started in 2012 and was characterised by demographic stagnation in connection with reduced migration inflows. This situation of slow population growth tends to be associated with the end of the process of urban dispersion. However, only after some time shall we be able to assess if such a process has really come to an end, while some of the latest trends have reopened the debate about the future prospects of urban dispersion in the BMR.

*3.2. The Choice of Living in a Suburban Residential Complex: Portraying the Actors of Residential Dispersion in the BMR*

An analysis of the information gathered by the survey "Mobility, Family Solidarity, and Citizenship in Metropolitan Regions 2005" allowed us to establish the characteristics of the population who had moved to suburban residential complexes. These were people who had changed their residence mostly after 1996; 34% of them coming from the city of Barcelona, and 40% coming from the rest of the BMR. Their profile was that of young people with a great potential for growth (between 25 and 45 years of age); 20.39% were aged under 15, and only 10% were older than 65 years. Predominantly,

they were in active employment, with both members of the couple working. Forty-five percent of those in work belonged to the categories of technicians, professionals, and managerial staff, and 56.8% were in medium–high socio-occupational groups. The prevalent family structure was that of a couple with underaged children (51.78%) living in a single-family house which was large (mean surface was 176 m$^2$), new (one-third of them had been built after 1985), and owned by the residents, but pending full payment (48.2%).

The appeal of dispersed areas was apparent in the reasons stated by the interviewees for moving to a suburban residential complex, which mostly had to do with the characteristics of the dwelling, the quality of life, nature, and the environment, and of lesser importance, with factors related to the actors' life trajectories, such as the creation or the extension of a family (Table 4).

**Table 4.** Reasons for moving to a suburban residential complex.

| Reason | % |
|---|---|
| Residential reasons (home and environment) | 54.1 |
| Reasons related to the dwelling | 31.6 |
| Desire to own one's home | 4.2 |
| Quality of life, environment, nature | 14.8 |
| Moving out of the city | 3.4 |
| Changes in the life cycle | 26.2 |
| Marriage or stable union | 10.9 |
| Family growth | 8.7 |
| Family reduction | 1.2 |
| Break-up of sentimental union | 2.1 |
| Retirement | 2.1 |
| Retirement of one's partner | 1.2 |
| Work-related reasons | 9.3 |
| Change of jobs | 3.9 |
| Partner's change of jobs | 1.6 |
| Closeness to (own/partner's) place of work | 3.7 |
| Health | 5.4 |
| Taking care of an elderly person | 1.8 |
| Health-related reasons | 3.6 |
| Other | 6.6 |
| Financial reasons | 1.3 |
| Total | 100 |

Source: Compiled by the authors based on the surveys "Mobility, Family Solidarity, and Citizenship in Metropolitan Regions 2005" and "Social Change and Urban Transformation Processes in a Context of Crisis in the Periphery of the BMR 2017".

The most significant reasons were those connected to residential factors (54.1%) and, in particular, the conditions of the house (31.6%): its price, the fact of being a single-family house or a newly built one, its surface area, its location in a natural environment and the quality of life (14.8%), or the desire to become the owner of one's residence (4.2%). Changes in people's life trajectories also affected the decision to live in a dispersed area (26.2%). In this respect, getting established as a couple (10.9%) or expanding the family (8.7%) were the most frequently stated reasons in this category. On the contrary, work-related motives did not play a very significant role, amounting to only 9.3% of the stated reasons. It must be noted that the interviewees prioritised reasons related to the appeal of low-density suburban areas over the rejection of denser areas. Thus, contrary to the North-American case [41], there is no mention of criminality, greater dangers, or higher pollution rates as reasons for migration.

This pattern of migratory behaviour produces a sex and age structure which both contrasts with and complements that of compact municipalities (Figure 5), which are actually the places feeding

the migration flows towards dispersed municipalities. Thus, we find a population which is quite rejuvenated, with a strong presence of residents between the ages of 40 and 59 and little pressure from elderly groups.

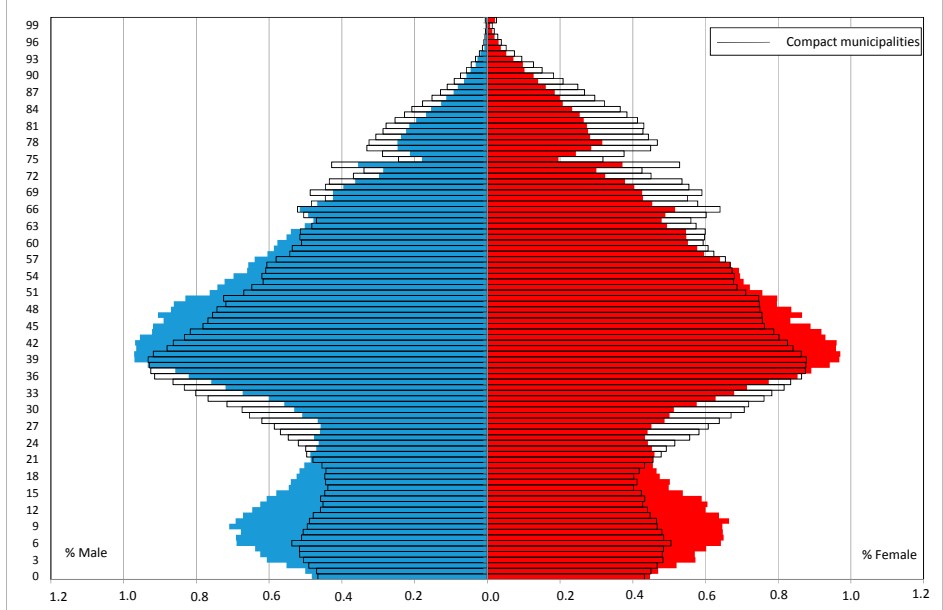

**Figure 5.** Sex and age structure in BMR dispersed municipalities, 2015. Source: Compiled by the authors based on *Padrón Continuo* (Continuous Register), 1 January 2015, INE.

When we consider the intrametropolitan migration profiles of different age groups, we appreciate the key role played by the strong appeal of dispersed residential areas to those between 25 and 44 years old, ages at which net migrations rates are the highest (Figure 6).

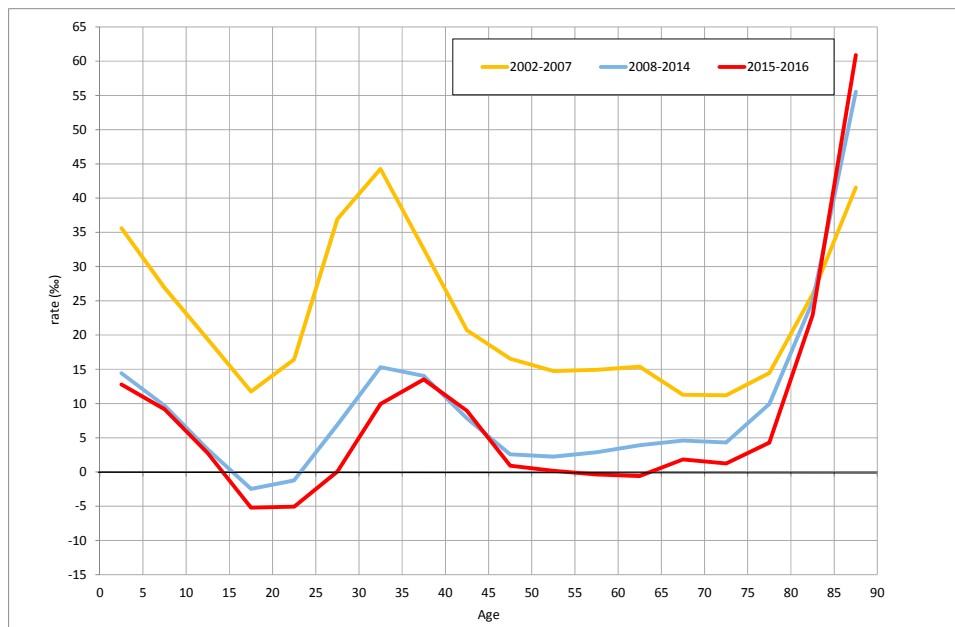

**Figure 6.** Net intrametropolitan migration rates by age in BMR dispersed municipalities (2002–2016). Source: Compiled by the authors based on data from the *Padrón continuo* (Continuous Register) 1996–2017 and the *Estadística de Variaciones Residenciales* (Residential Change Statistics) 1996–2016, microdata file, INE.

In this case, we must highlight the fact that this age pattern repeats itself over time, even when the effects of the economic crisis pull down the rates of migration towards dispersed destinations. Therefore, it is obvious that dispersed municipalities retain their appeal as places of residence for a specific target population, an appeal that may be boosted or curbed by the economic situation, but remains in place throughout our period of observation. Parallel to this, the mobility of adults has a pull on the mobility of both underaged young people, who migrated with their parents, and elderly people, who migrated in order to live closer to their children, who moved earlier in time, and thus moved in order to receive care from or provide care to them.

On the other hand, what we see is a pattern of residential appeal clearly segmented by age and socioeconomic category. This is a phenomenon which has been well described in the literature about migration focusing on the analysis of changes in residential preferences in relation with life stages [42,43].

### 3.3. Twelve Years Living in a Suburban Residential Complex: The Actors' Assessment

The last point we should like to stress in this study is the residents' assessment of, and satisfaction with, the suburban residential complex and housing, as revealed by our 2005 and 2017 surveys.

The answers to the questions regarding the level of residential satisfaction in our surveys display high values in relation to both the place of residence and the dwelling. In fact, the results obtained by the 2017 survey are even better than those of the 2005 one (Tables 5 and 6). Forty-five percent of the interviewees declared that the suburban residential complex where they lived had improved since 2005. Altogether, the average score obtained by the residential place of living in the survey was 4.2 out of 5, while people's satisfaction with their homes reached an average score of 4.5 out of 5. The interviewees were more critical about the town council administration of their residential areas, which they valued at 5.4 points out of 10.

**Table 5.** Evaluation of the residential area and the dwelling, 2017.

| Score | Suburban Residential Complex | Dwelling |
|---|---|---|
| 1 | 2.0 | 0.0 |
| 2 | 2.8 | 0.7 |
| 3 | 13.8 | 5.2 |
| 4 | 26.3 | 26.7 |
| 5 | 44.8 | 67.5 |
| Total | 100.0 | 100.0 |
| Average Score | 4.2 | 4.6 |

Source: Compiled by the authors based on the survey "Social Change and Urban Transformation Processes in a Context of Crisis in the Periphery of the BMR 2017".

**Table 6.** Perception of the town council actions in the residential area, 2005–2017.

| Score | 2005 | 2017 | Diference |
|---|---|---|---|
| 0 | 13.3 | 7.5 | −5.8 |
| 1 | 4.0 | 2.2 | −1.8 |
| 2 | 6.0 | 4.2 | −1.8 |
| 3 | 7.7 | 7.0 | −0.7 |
| 4 | 10.3 | 7.7 | −2.6 |
| 5 | 19.8 | 19.3 | −0.5 |
| 6 | 13.0 | 13.7 | 0.7 |
| 7 | 9.0 | 17.8 | 8.8 |
| 8 | 10.3 | 12.8 | 2.5 |
| 9 | 1.0 | 2.8 | 1.8 |
| 10 | 2.3 | 4.0 | 1.7 |
| Total | 100.0 | 100.0 | 0.0 |

Source: Compiled by the authors based on the surveys "Mobility, Family Solidarity, and Citizenship in Metropolitan Regions 2005" and "Social Change and Urban Transformation Processes in a Context of Crisis in the Periphery of the BMR 2017".

This information is especially significant for our investigation for two reasons: Firstly, because it tells us about the impact that the economic crisis had on the families that chose to move to this type of residential area. Thus, a drop in household income or in the funds allocated for the provision of services by public administrations—as a consequence of budget cuts imposed by the economic crisis—might give rise to a less positive evaluation of a residential choice that was made years before, in the context of economic welfare. Secondly, because the existence of high levels of dissatisfaction would point to the possible failure or rejection of life in dispersed residential areas, and this might lead to a new change of residence.

## 4. Discussion

The results presented in the previous section take us to the discussion of the future of dispersed residential areas as derived from their recent trajectory and the information gathered by our research up to this point.

First of all, it must be highlighted that despite the slowing down of the development of dispersed urbanism imposed by the economic crisis, the available intrametropolitan mobility data reveal that dispersed municipalities have not completely lost their appeal. They still display net migration rates higher than the rest of the urban typologies considered for the BMR. For this reason, it seems that its present situation is more of a phase of lethargy, while waiting for the evolution of the economy and the credit market, than one of disappearance of the residential model. This is the key to understanding the increase of net migration rates in 2015 and 2016, when the economic recovery started to be timidly revealed in some economic sectors. As can be seen in Figure 4, in those two years, the previous trends pointing to a sustained reduction of net migration rates were reversed. It is significant that this recent increase in net migration in dispersed municipalities coincided with a rise in the loss of population of more compact areas in the BMR due to intrametropolitan out-migration. This change of trend did not only coincide with the beginning of the economic recovery, but also with another phenomenon that took place at the same time, namely, the renewed increase in the price of housing in large metropolitan cities—both in the sale and rental markets. In the city of Barcelona, as it happened in other cities [44], the purchasing price of a newly built home rose by 20.05% between 2015 and 2017 (Table 7). As for rent prices, they increased by 19.36% [45]. This situation, which some commentators consider as a new real estate bubble, intensifies the process of population expulsion from the denser cities and stands in the way of the arrival or the return of potential residents to them.

**Table 7.** Evolution of the sale price of housing in Barcelona (2004–2017).

| Year | New Housing | | Used Housing | | Total Housing | |
|---|---|---|---|---|---|---|
| | Value (€/m$^2$) | Change (%) | Value (€/m$^2$) | Change (%) | Value (€/m$^2$) | Change (%) |
| 2004 | 3336 | - | 2986 | - | 3079 | - |
| 2005 | 3708 | 11.1 | 3782 | 26.7 | 3758 | 22.1 |
| 2006 | 4452 | 20.1 | 4296 | 13.6 | 4349 | 15.7 |
| 2007 | 5009 | 12.5 | 4505 | 4.9 | 4622 | 6.3 |
| 2008 | 5144 | 2.7 | 4235 | −6.0 | 4464 | −3.4 |
| 2009 | 4264 | −17.1 | 3643 | −14.0 | 3773 | −15.5 |
| 2010 | 4259 | −0.1 | 3577 | −1.8 | 3745 | −0.8 |
| 2011 | 4276 | 0.4 | 3327 | −7.0 | 3559 | -5.0 |
| 2012 | 3109 | −27.3 | 2904 | −12.7 | 2946 | −17.2 |
| 2013 | 3197 | 2.9 | 2628 | −9.5 | 2719 | −7.7 |
| 2014 | 3116 | −2.5 | 2705 | 3.0 | 2754 | 1.3 |
| 2015 | 3237 | 3.9 | 2934 | 8.4 | 2971 | 7.8 |
| 2016 | 3850 | 18.9 | 3167 | 7.9 | 3238 | 9.0 |
| 2017 | 4048 | 5.2 | 3714 | 17.3 | 3746 | 15.7 |

Source: Generalitat de Catalunya Departament de Territori i Sostenibilitat (Department of Territory and Sustainability) (2018). *Informe sobre el sector de l'habitatge a Catalunya* (Report on the Housing Sector in Catalonia) 2017 [44]

Given this state of affairs, the supply of housing in dispersed municipalities may gain prominence once more and provide a choice for intrametropolitan in-migration again. It will be necessary to stay attentive to the evolution of the metropolitan supply of housing, knowing that it is a type of market that generates dynamics which put different urban areas in relation to one another, as has happened before. Once again, the general evolution of the country's economy and the credit policies of its financial institutions will play a key role, as Lomax and Stillwell pointed out for the case of the United Kingdom [46].

The renewed increase in the price of housing in larger cities might produce a renewed dynamisation of the real estate market by encouraging the sale or renting out of homes; at the same time, this could reactivate migration flows towards dispersed municipalities, which boast a large supply of housing and a great potential for growth. In the same way, an exorbitant rise in the price of housing—both in rents and purchasing prices—might give rise to new family strategies such as the transference of second homes to their children (or the moving of parents to such second residences) in order to make it possible for the younger members of the family—victims of the recent skyrocketing of housing prices—to become emancipated.

On the other hand, it is obvious that the overrepresentation of some age groups will determine the municipal agenda for the planning of services and infrastructure. Besides this, the movement of these age groups up the population pyramid will also reshape the demands that the different administrations—especially the local ones—will have to satisfy (Figure 5). Moreover, if we examine the behaviour of migratory trends for the eldest age groups, we still observe net gains during all the periods under consideration and with similar intensity across time. This is another factor that could sensibly modify the demand for services. The residential strategies of the elderly display certain features of their own, due to the fact that mobility is here associated with preparation for old age. Thus, to the appeal for residential quality we must add other factors, such as proximity to family or friends and to services, and the rejection of excessive dependence on private vehicles for transportation [47]. Therefore, prima facie, the appeal of dipersed municipalities for elderly people that is evident in the case of the BMR would hardly agree with the behaviour stipulated by some theoretical frameworks.

The prevalent theoretical models often point to people's aging and entering an empty-nest life stage as factors leading to the relocation of populations. In our case, however, the results of the surveys show little intention of people to change places of residence, move again, and/or return to their previous areas of residence. Thus, there is a determination to become old in the same suburban residential complex: only 16% of the interviewees seemed to have taken into consideration a change of residence, and paradoxically, individuals aged between 40 and 54 years seemed more likely than elderly people to embark on a new migration process. Only a small minority declared their will to change homes in search of a dwelling with different characteristics: smaller, requiring less maintenance, and located in an area where there is not so much dependence on private means of transport. What our information reveals is a will to get old in areas of dispersed urbanism.

If one of the dimensions exposed by previous research was the weakness and shortage of family and support networks in environments of dispersed urbanism [48], the information gathered in our 2017 survey makes it clear that with the passing of time, people have managed to weave social networks that act as fixing factors and make residents stay in the suburban areas where they have spent their latest years. Even though emancipated children do not reproduce their parents' residential model (only 23% of them stay in the same municipality where their parents' suburban area of residence is located), our study reveals the beginning of the formation of the so-called family 'entourage' [49], which would reinforce people's attachment to their place of residence. This strategy of residential relocation in search of closer proximity to other family members does not only take the form of staying near the parents' home, but also accounts for the high in-migration rates of elderly people, who look for their children's vicinity in order to develop strategies of intergenerational solidarity.

On the whole, the worsening of economic indicators does not affect people living in dispersed areas in a more acute manner, despite the special features associated to this style of living: relative

isolation, distant services, extra costs, and greater weakness of support networks. In this respect, the significant presence of middle- and upper-class households may account for the less dramatic impact of the economic crisis and the absence of situations of serious social degradation, situations that were feared at some point in time. Due to the lack of a tradition of living in dispered areas in Spain, the gradual evolution of the residential choices of families is observed with even greater interest, as those who moved to low-density areas in the 1980s were pioneers in displaying the effects of changes in life circumstances and in the business cycle on residential strategies involving migratory exchanges between dispersed and compact areas.

## 5. Conclusions: A Look into the Future

Nowadays, the key question concerning dispersed urbanism is what its future propects are in the middle and long term. As we have seen in the previous sections, residential sprawl occurred in Spain in the context of economic growth, accompanied by a series of other factors which facilitated its swift expansion. The setting in of the economic crisis brought such rapid expansion to a sudden halt, which was interpreted as the end of a process. This verdict was founded on the extra costs of living in dispersed areas, which would act as a deterrent for new residents and would favour a retreat to compact residential areas, and on the hardening of the requirements to access a mortgage loan, which would be the cause of putting off or giving up the purchase of a home. However, such views were not completely right in their predictions.

Despite the time already elapsed after their arrival in a dispersed residential area—all the interviewees had been living in a suburban area for at least 10 years—the hard impact of the economic crisis, and the expenses associated with living in such an environment, the level of satisfaction of the people who took part in our surveys was in 2005, and was still in 2017, very high. Thus, the crisis did not alter the residents' assessment of the option of living in a dispersed residential area.

As we have verified, the economic crisis curbed in-migration and limited the age range in which net migration took a positive value. Nevertheless, out-migration did not increase, and it even went down in the reported years. The conclusion is that dispersed areas retained their appeal in the stages of creation and expansion of households, as theoretical models point out. For this reason, an effective economic recovery and a renewed rise in the price of housing in denser cities—with Barcelona as an outstanding exponent of such an evolution—may contribute to an upturn in the popularity of the dispersed residential model, which nowadays could be considered to be in a 'lethargic' stage, waiting for certain factors to concur and reactivate it. Such reactivation would mostly involve the most affluent socioeconomic groups, being the only ones fulfilling the requirements to gain access to a mortgage loan or who might not even need one, unless we witness an easing of the conditions to obtain a mortgage. As a consequence, this type of residential mobility could be restricted to high-income households, which would result in the concurrence of processes of residential dispersion and social segregation.

Finally, neither the economic crisis nor the passing of time or the change of life-cycle stage have altered the indexes of residential satisfaction of our interviewees. There is a sustained highly positive appraisal of both the dwelling and the suburban areas of residence, while the presence of projects of relocation and of families waiting to sell their houses to move out of the area is scarce.

In conclusion, there is a demand for the lifestyle that dispersed urbanism represents, and in parallel, there does not appear to be a process of out-migration or strong rejection of such a residential type. Everything seems to suggest that dispersed urbanism is here to stay.

**Author Contributions:** A.G.-C. is a geographer specialising in residential mobility and has been in charge of drafting the sections concerning migration processes. C.L.-V. is a Sociology PhD and so her contribution has focused mainly on the social aspects dealt with in the article. Both authors have conducted joint studies of the processes of urban dispersion since 2004, with a special focus on the definition and measurement of dispersion and its demographic impact.

**Funding:** This piece of research has been carried out in the framework of the R + D + i project entitled "Social Change and Processes of Urban Transformation in a Context of Crisis in the Barcelona Metropolitan

Region", reference number CSO2013-48075-C2-1-R, funded by the Spanish Ministry of Economy, Industry and Competitiveness, State Research Agency, Spain.

**Acknowledgments:** The authors are grateful to the members of the collaborating team in the project "Social Change and Processes of Urban Transformation in a Context of Crisis in the Barcelona Metropolitan Region".

**Conflicts of Interest:** The authors declare no conflict of interest.

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
