# Peer review of "The Impact of Economic Crisis in Areas of Sprawl in Spanish Cities"

_urbansci, doi:10.3390/urbansci2040113_

Round 1

Reviewer 1 Report

The paper discusses the case study of Barcelona Metropolitan Region, which is presented as being significative to analyse the impact of economic crisis on the processes of urban growth in Spain. In my opinion, the application could be of interest for the readers of the journal. However, some revision must be done before considering the manuscript for publishing.

1) My main concerns are related to the relevance of the application for researchers who are not specifically interested into the Spanish case study (nor on the Barcelona’s one). Why do author think this paper could be of interest for other researchers? Moreover, authors scarcely discussed the relevant bibliography in the field of land use change and urban growth. Most of the reference are related to the Spanish situation, indeed. Isn’t urban growth a relevant issue also for other countries? I was also expecting a wide discussion on the similarities among Spain and other Mediterranean countries.

To reinforce the paper under these points of view, I would suggest to revise some of the literature in the research field of land use change and urban growth. I suggest as starting point the following relevant papers, but I expect the authors to reinforce their statements using even more references:

- N. Ramankutty and O. T. Coomes, “Land-use regime shifts: an analytical framework and agenda for future land-use research,” Ecol. Soc., vol. 21, no. 2, p. art1, 2016.

- F. Amato, M. Tonini, B. Murgante, and M. Kanevski, “Fuzzy definition of Rural Urban Interface: An application based on land use change scenarios in Portugal,” Environ. Model. Softw., vol. 104, 2018.

2) The structure of the paper must be revised. Generally speaking, the methodology is not well presented in the corresponding section, and the distinction between result and discussion is not clearly understandable. Lines 107-127 probably belong to the introduction, not on the methodology. The results section now contains also the discussion one, so I do not understand why authors called the last section “discussion and conclusion”.

Author Response

 Review Report (Reviewer 1)

We really appreciate the revision and suggestions for improvement our article. We have modified and improved the suggested aspects as follow:

1)       Relevance of the application for researchers who are not specifically interested into the Spanish case study (nor on the Barcelona’s one). Why do author think this paper could be of interest for other researchers?

We have added the explanation on why this paper could be of interest to other research, even if they are not interested in Spanish case study:

The BMR is a good example of a metropolitan region where dispersed urban developement set in late and grew at an accelerated pace. Inititally, the territories were not prepared to respond to the new situations generated by the dispersed model and, thus, their adaptation to their new changing realities could only take place simultaneously to the implementation of the new urban model. By studying these features of the BMR case, the present article attempts to contribute some empirical evidence to the debates about the specificities of the establishment of dispersed urbanism outside the Anglo-Saxon regions, in line with the works of Couch et al. [18] or Richardson and Bae[2]. More particularly, it is our aim to help devise a framework which is useful to specify the role of the economic crisis in processes of urban dispersion, according to the findings of Salvati[5] o Cho et al.[19], whose research discloses the undeniable effects of the economic crisis, which not only puts an end to construction, but also to trends of suburban development in a territory.

Also, the present article argues for the need of incorporating the perspective of demographic studies into morphological and urbanisation studies, with demographic studies focusing on specifying the elements responsible for residential dispersion, such as residential mobility or the different directions of this mobility depending on people’s stage in life. To this end, our research combines information coming from official statistics on residential mobility in Spain with data coming from two surveys of people who moved to live in a dispersed setting. These surveys do not only inform us of the causes behind people’s migration, but they also provide us with the interviewees’ assessment of their change of residence and information about their future migration projects. Often, as Champion[21], Tyrell and Kraftl[22], or Coulter et al.[23] suggest, lack of data leads to analyses of the socio-economic factors which determine urban sprawl which only take into consideration the moment when people’s migration take place (e.g., the analyses of Weilemmann et al. [20] for Switzerland, or Salvati[5] for Athens), without taking into account people’s whole migratory course of life. This renders such analyses unable to tell us anything about future migration prospects. As Coulter et al.[23] state, the point here is to re-think residential mobility “linking lives through time and space.”

Nowadays, low-density urbanism occupies a prominent place in territorial planning, since the changes in land use that it entails make it a relevant factor to be considered when planning both the present of urban development and, above all, its future prospects, as it is aptly observed by Ramankutty and Coomes[24].

2)       Isn’t urban growth a relevant issue also for other countries? I was also expecting a wide discussion on the similarities among Spain and other Mediterranean countries.

We have added more references and explanations about disperse process in Mediterranean countries and similarities and differences among Spain and other Mediterranean countries.

Català et al.[4] point out the existence of common elements in the processes of urban dispersion undergone by the European Mediterranean countries. They specifically identify the following shared features: very late occurrence of the sprawl phenomenon; extreme urban densification in previous stages; great vitality of central cities; and a high presence of small and medium-sized cities in the country’s urban system. They also point to the existence of three overlapping urban models: the traditional compact model, the dispersed model of the new residential areas, and the mixed land uses exemplified by industrial and commercial activities that tend to occupy large land containers. Salvati[5] describes this last model as “hybrid”, underscoring its importance in order to understand the particularities of Mediterranean Europe.

Despite all these shared aspects, Carlutti et al.[6] have recently centred on the differences between individual Mediterranean countries to reach the conclusion that it is not possible to speak of a single model of urban dispersion in the Mediterranean Europe and it is necessary to take into account both regional and local factors to understand the specific evolution of each one of the countries in the area.

Spanish urban regions are a good example of the situation described by the Washington Post article, as well as of the Mediterranean model of urban dispersion. On comparing the sprawling processes in twenty-six European countries in the 1990s and the first half of the 2000s, Siedentop and Fina[7] concluded that Ireland, Portugal and Spain were the European countries which had experienced the the greatest sprawl. After several decades of urban growth based on a model of compact urban development, at the turn of the century Spanish cities began to change their patterns of urban expansion. On the one hand, they experienced an accelerated tendency towards suburbanization, characterized by a loss of both population and compactness in the largest municipalities and the growth of their peripheral areas, which became more extensive[8, 9]. On the other hand, a new model of dispersed city was established, a model which had been rarely seen before and scarcely relevant among Spanish urban territorial trends, besides being mostly associated with second homes. And it is this last fact that constitutes a distinctive factor of the Spanish process of urban dispersion [10]. Thus, in the 1960s, Spain saw the development of areas of residential expansion at a short distance from the cities, areas which were mostly devoted to second homes and which in more recent stages would act as the germ of subsequent dispersed urban development. According to Burriel[11], this was a phenomenon unmatched in the rest of Europe and it turned out to be a crucial factor to be taken into account if we wish to understand the expansion of dispersed urbanism in Spain at the end of the 20th century.

3)       The structure of the paper must be revised. Generally speaking, the methodology is not well presented in the corresponding section, Lines 107-127 probably belong to the introduction, not on the methodology.

Following the suggestion done by reviewer 3, we have added subtitles to point 2, dividing the article in:  methodological approach, methods, data analysis in the methodology.  We have written an introduction to point 2, linking introduction and methodology.

4)       The distinction between result and discussion is not clearly understandable.The results section now contains also the discussion one, so I do not understand why authors called the last section “discussion and conclusion”.

We have separated discussion and conclusion and we have reorganised “results”, “discussion” and “conclusion”

New text included is written in red color in the paper.  Highlighted in yelow, it’s possible to find text that has been moved. English language and style have been checked.

Reviewer 2 Report

Dear authors, 

Thank you for sending this paper for revision. I enjoyed reading your work. 

My observations are minor: 

1) I think the term 'urban sprawl' is misused. Indeed (and instead), you keep using the term 'disperse urbanism' as appropriate to identify the pattern of urban development you are describing/analysing. I think 'disperse urbanism' is much better in comparison to 'urban sprawl'. 

The term 'urban sprawl' is a very technical one - born as part of the Anglo-Saxon lexicon in Geography/Planning - and it is used to describe patterns of urban growth that are clearly characterised by morphological but also functional factors related to land fragmentation, vehicle miles travelled, morphological dispersion, land-plot subdivision, and others including density, nuclearity, continuity, and others. These factors are already used to calculate the 'sprawl index' and thanks to this, it is possible to assert that not all the peri-urban landscapes of city regions describe an instance of 'urban sprawl. That means that in the same city, some areas can be possible to be identified as sprawling ones and some others do not. So, there is distinction between urban sprawl and dispersed suburbanisation, metropolitan fragmentation, etc. I assume that the literal translation from English to Spanish of 'urban sprawl' is similar to 'disperse urbanisation', but again, strictly speaking the term 'urban sprawl' is not the same and has gained a meaning in its own right, thus, not possible to be literally translated.  

So, I suggest keeping the term you already used: disperse urbanism. This seems much more appropriate to what you are analysing. (Indeed, the concept 'urbanism' is also different from 'urbanisation' as the term 'urbanism' is the Latin-root mixed of 'humanism' and urbanisation' = 'urbanism'). So, please check this out and expand the understanding of this terms at the beginning (in the Introduction). 

2) There are 7 key words. Is that OK according to the Journal norms? Please check this out.

3) The word 'Register' (line 155, 156, captions of Table 1) seems to be 'Registry' instead. Please check the English out. 

4) It might be helpful to add some subtitles in the methodology (Methodological approach, methods, data analysis). These sub-sections are implicitly there already, but can be helpful for the readers if the methodology can be a bit splitted up to keep clarity of its different contents. It might be helpful considering the (somehow) length of the Methodology. 

Author Response

 Review Report (Reviewer 2)

We really appreciate the revision and suggestions for improvement our article. We have modified and improved the suggested aspects as follow:

1) I think the term 'urban sprawl' is misused. Indeed (and instead), you keep using the term 'disperse urbanism' as appropriate to identify the pattern of urban development you are describing/analysing. I think 'disperse urbanism' is much better in comparison to 'urban sprawl'. 

We have changed “sprawl” by “disperse urbanism”

2) There are 7 key words. Is that OK according to the Journal norms? Please check this out.

Journal template indicates: “Keywords: keyword 1; keyword 2; keyword 3 (List three to ten pertinent keywords specific to the article; yet reasonably common within the subject discipline.)”

3) The word 'Register' (line 155, 156, captions of Table 1) seems to be 'Registry' instead. Please check the English out. 

We have not changed the word Register by Registry. We have checked out the official nomenclature used by Spanish Statistical Institute and word concept used is Register (http://www.ine.es/dyngs/INEbase/en/categoria.htm?c=Estadistica_P&cid=1254734710990)

4) It might be helpful to add some subtitles in the methodology (Methodological approach, methods, data analysis). These sub-sections are implicitly there already, but can be helpful for the readers if the methodology can be a bit splitted up to keep clarity of its different contents. It might be helpful considering the (somehow) length of the Methodology. 

We have added subtitles in the methodology: methodological approach, methods, data analysis. It helps to readers to appreciate different contents. We have written an introduction to point 2, linking introduction and methodology.

2. Materials and Methods

In this section, we shall deal with three basic issues that constitute the departing point of our research. In the first place, we will centre on methodological questions to reflect about both the theoretical and the empirical difficulties when it comes to defining dispersed urbanism. As we shall see, although it is true that dispersed urbanism has been the subject of a good number of both theoretical and applied studies, its definition and measurement has important limitations, independent of whether we consider the Spanish case or that of other countries. Secondly, we shall introduce the methodology employed in our research, which was based on a classification of municipalities specifically designed to identify those with a strong presence of dispersed urbanism. Finally, we shall proceed to the description of the sources used in our empirical analysis.

New text included is written in red color in the paper.  Highlighted in yelow, it’s possible to find text that has been moved. English language and style have been checked.

Changes included in the text are written in red color in the paper.

Reviewer 3 Report

This work is very interesting and i believe it is an appealing reading for the audience of Urban Science. The paper’s structure is adequate, although some restructuring is needed for the last paragraphs. It focuses on the many interrogatives dealing on the one hand with spatial patterns of urbanization in a time of crisis, while on the other hand investigating also the social characterization. The manuscript aims at diachronically analysing expansion of urbanization in Barcelona (BMR), and to interpret differential patterns of urbanization according to different explanatory factors. I frankly enjoyed the paper. However I believe that some minor caveats still remains. For example, literature review is strongly “Spain based” and neglects to consider important contribution to the discipline around urban sprawl that come from elsewhere. Furthermore, authors hypothesize an alternation between sprawl dynamics (which they consider to possibly be in a lethargic state) and compact growth, but they neglected to consider authors that have already proficiently investigated such dynamics, and whose contribution would definitely improve the significance of the results provided by authors. Nevertheless, also some of the analysis and data used are not described with enough details. Therefore, It is my modest opinion, that the manuscript would greatly benefit from a revision taking into consideration the limitation listed below:

1) line 145-149. Thresholds for different residential density based on population have been used. However, the appropriateness of these thresholds is not discussed. Where these numbers come from? I mean, i am not questioning that authors should have used different thresholds, but the fact that they are not considering that different thresholds could be used. Therefore, either authors use references to support and justify these values, or they provide a numerical analysis or method eliciting such results, or they provide a sensitivity analysis eliciting the robustness of the values used.

2) line 179-198. This entire paragraph is fundamental to correctly understand the rationale of the paper. However authors mention that data have been taken from some surveys they have conducted in the past. This is not an adequate description. In fact i believe authors should either better support through references the afore mentioned surveys, or use supplementary material to better describe how surveys have been carried out, results of surveys, potential biases of these surveys, and main contribution to the present work. Some statistical evidence would be beneficial also to the paragraph itself.

3) line 229-243 The part b) is expressed confusedly and requires a rewriting. Try to use shorter sentences to improve readability. Furthermore, the identification of these 5 strategies is not well explained. There are no graphs for example linking prices trend with urbanization degree. As a matter of fact the function identified in figure 2 are pretty much similar, although having different coefficients. Probably, highlighting onto the graph itself the breaking points and label them could be helpful to the reader to immediately identify the two periods authors are talking about

4) figure 3 is also not so clear, try to make more clear that we are looking at the trend over time from 1996 to 2016 for 5 different types of urbanization. Arrows may induce reader to think that this is a progressive dynamic, which is not, because you simply ordered types form compact to dispersed, but in the geographical space this may not be always the case. Secondly if it is simply the comparison between 1996-2007-2016 that you are interested, maybe a simpler graph with bars or line for the 5 types could be easier to read

5) This is a caveat that authors should somehow try to address or discuss. Table 4 is very interesting; however, this info should be more strongly linked to the urbanization and net migration graphs presented before. But in here this info is given as a n average for the whole Barcelona, while it would definitely more helpful, and support conclusion more robustly, if this info was provided per type of urban density

6) in general paragraph 3 and 4, results and discussion and conclusion, need to be restructured. So far there is too much of discussion in the presentation of results. While results should only describe what ca be said with the elaboration and manipulation of data. Discussion throughout the two paragraphs is good, but deserves a single paragraph eliciting the problems and speculations that can be done at the aid of this data. Conclusion should focus on summarizing take home messages and posing further questions for the future. Therefore the suggestion is to make three separate paragraphs from these two. Furthermore, be sure to discuss results in light of the points given in the introduction while stating the research questions, and also in regards of the missing bibliography.

7) The aim of the paper is to describe sprawled urban patterns in light of economic crisis. The manuscript focuses on the one hand on spatial urban pattern and on the other hand on socio economic factors. However for both themes, in my modest opinion, there are important lacks in bibliographic references, while very well is treated the part dealing with migration dynamics. As a matter of fact there is a bulk of literature that investigated sprawl in a similar way, and that somehow should be considered in the introduction and discussion. Doing so would entangle findings of this work more pertinently in the corpus of significant literature dealing with sprawl, and making this work more relevant also for other authors.

In particular I find lacks for what concerns the spatial dimension. As a matter of fact, I wonder whether a simple land density function (based on built up) would differ much from the density function used here. Furthermore, this land density function (as all land density) are biased by the total amount of land available (as in Shannon spatial entropy theory, or Moran dispersion index), changing the initial window of land considered will change also density results. But the aim here is not to focus on density itself, rather to distinguish macro pattern of urban growth dynamics (sprawl vs coalescence). In this regard, for example see:

·         Jochen A.G. Jaeger, Christian Schwick 2014, Improving the measurement of urban sprawl: Weighted Urban Proliferation (WUP) and its application to Switzerland, Ecological Indicators, Volume 38;

This bit of additional literature gives a panorama of alternatives for investigating land density, and more specifically sprawl, which do not have some of the bias of the method used. I am not saying authors should change the method of analysis, but at least be aware and acknowledge that other methods exist, which overcome limitation of the method chosen, and that maybe a sensitivity analysis discussing/comparing these methods should be done to improve the soundness of the analysis. Or at least explain why they used a simpler method while such other exist. For example data limitation can be an explanation, or that authors wanted the focus for this study to be more centred on population rather than on spatial structures.

Furthermore, as said in the beginning, and also stated by authors in line 107-111,  the literature about urban expansion dynamic and measurements is rich, and features also several theories about the evolution of urban dynamics through times that here authors seem to completely neglect to consider (see for example:

·         Martellozzo and Clarke 2013. Urban Sprawl and the Quantification of Spatial Dispersion. In Geographic Information Analysis for Sustainable Development and Economic Planning: New Technologies. 129-142. IGI - Global

These theories and the results proposed are very much relevant, in my modest opinion, to better frame the work hereby presented. For example the oscillation between period in which urbanization is influence more by coalesced growth, and vice versa, the indices developed to capture this sprawl degree or the fractal character of urbanization dynamics are something that would implement and incorporate more evidently the work hereby proposed with existing literature on the topic. Plus is something that authors state directly in the beginning while referring to the “lethargic” state of dispersion. Very appealing definition, in my opinion, that would benefit for a contextualization with the other studies on the topic.

Nevertheless, there are also some lacks for what concerns the explanatory variables used in literature to interpret and describe urbanization, which could implement the work of authors. In this regard see:

·         Weilenmann B., Seidl I., Schulz T., The socio-economic determinants of urban sprawl between 1980 and2010 in Switzerland, Landscape and Urban Planning 157 (2017) 468–482

Author Response

Review Report (Reviewer 3)

We really appreciate the revision and suggestions for improvement our article. We have modified and improved the suggested aspects as follow:

1)       line 145-149. Thresholds for different residential density based on population have been used. However, the appropriateness of these thresholds is not discussed. Where these numbers come from? I mean, i am not questioning that authors should have used different thresholds, but the fact that they are not considering that different thresholds could be used. Therefore, either authors use references to support and justify these values, or they provide a numerical analysis or method eliciting such results, or they provide a sensitivity analysis eliciting the robustness of the values used.

We identified five types of municipalities, categorizing as ‘dispersed’ those situated in the first and second quintiles of the net-density distribution, i.e., with less than 140 inhabitants/hectare in 2015. The percentage of land allocated to detached/semi-detached houses was used to distinguish between dispersed and highly dispersed areas as internal categories of urban dispersion (Table 1). According to 2015 data, in the BMR there are 106 municipalities (64.7 percent of all) which satisfy the criteria for dispersed areas, amounting to 14.2 of the BMR population.

We have modified table 1

2)       line 179-198. This entire paragraph is fundamental to correctly understand the rationale of the paper. However authors mention that data have been taken from some surveys they have conducted in the past. This is not an adequate description. In fact i believe authors should either better support through references the afore mentioned surveys, or use supplementary material to better describe how surveys have been carried out, results of surveys, potential biases of these surveys, and main contribution to the present work. Some statistical evidence would be beneficial also to the paragraph itself.

We have added some extra information about surveys:

Additionally, as a complement to this preliminary observation of our object of study, we used the results of two surveys that we ourselves carried out in 2005 and 2017. Both of them had been designed to gather information about the living standards of people residing in dispersed areas in the BMR. One section of the questionaire was aimed at getting information about people’s migratory experiences. Here, it must be taken into account that, in 2005, all the interviewees had changed their municipality of residence at least once in their life time. More than 90 percent of them had lived in a compact area at some point and, thus, were able to compare both lifestyles. Also, all the interviewees had been living in a sprawl area for over 12 years, and so they had experienced the effects of the economic crisis while living in this type of residential area. Contrary to the basic general information provided by the Municipal Register or the EVR (which only include data about sex, age, place of birth and nationality), our surveys allowed us to complete the socio-demographic profiles of the interviewees (profession, employment status, household income, etc.) and gave us information about their motives, assessment of their residential area and future migratory projects, all of which was highly valuable to achieve the goals of our research.

Technical data were included in the previous version.

Results of surveys are use on “Results” epigraph.

3)       line 229-243 The part b) is expressed confusedly and requires a rewriting. Try to use shorter sentences to improve readability. Furthermore, the identification of these 5 strategies is not well explained. There are no graphs for example linking prices trend with urbanization degree. As a matter of fact the function identified in figure 2 are pretty much similar, although having different coefficients. Probably, highlighting onto the graph itself the breaking points and label them could be helpful to the reader to immediately identify the two periods authors are talking about

We have rewritten part b) (line 229-243 in the original version) with shorter sentences.

4)       figure 3 is also not so clear, try to make more clear that we are looking at the trend over time from 1996 to 2016 for 5 different types of urbanization. Arrows may induce reader to think that this is a progressive dynamic, which is not, because you simply ordered types form compact to dispersed, but in the geographical space this may not be always the case. Secondly if it is simply the comparison between 1996-2007-2016 that you are interested, maybe a simpler graph with bars or line for the 5 types could be easier to read

We have redrawn the figure 3

5)       This is a caveat that authors should somehow try to address or discuss. Table 4 is very interesting; however, this info should be more strongly linked to the urbanization and net migration graphs presented before. But in here this info is given as a n average for the whole Barcelona, while it would definitely more helpful, and support conclusion more robustly, if this info was provided per type of urban density

Evolution of the sale Price of housing is available for municipalities with more than 5,000 inhabitants. That means is not possible to re-built urban typology used in the classification.

6)       in general paragraph 3 and 4, results and discussion and conclusion, need to be restructured. So far there is too much of discussion in the presentation of results. While results should only describe what ca be said with the elaboration and manipulation of data. Discussion throughout the two paragraphs is good, but deserves a single paragraph eliciting the problems and speculations that can be done at the aid of this data. Conclusion should focus on summarizing take home messages and posing further questions for the future. Therefore the suggestion is to make three separate paragraphs from these two. Furthermore, be sure to discuss results in light of the points given in the introduction while stating the research questions, and also in regards of the missing bibliography.

We have separated discussion and conclusion and we have reorganised “results”, “discussion” and “conclusion”

7)       lacks in bibliographic references, in particular I find lacks for what concerns the spatial dimension.

We have added more bibliographic references, overall concerning to spatial dimension.

Multidimensional measurements of urban sprawl are frequent too, generally using GIS tools. Jaeger and Schwick[32] propose to use Weighted Urban Proliferation (WUP) to consider the area built over in a given landscape (amount of built-up area), the dispersion of the built-up area and the land uptake of the built-up area per inhabitant or job. Weillenmann et al.[20] consider that built-up areas may spill across administrative borders, with neighbouring municipalities sharing a similar employment market and mobility infrastructure and requiring coordination when it comes to spatial planning. All these sophisticated methodologies undoubtedly feed an interesting debate; however, the most important thing to do before making a decision regarding the definition of dispersed urbanism to be used in a particular study is to establish the research objectives and to evaluate if the statistical data needed are available.

In our case, the main objective revolved around demographic trends rather than spatial structures. Moreover, in the Spanish case, we came across a serious limitation concerning the potential sources of information available to researchers, namely: there exists no exact correspondence between the various administrative and statistical divisions and dispersed residential areas, especially when it comes to suburban residential complexes. Thus, it was not always feasible for us to reconstruct the socio-demographic traits of such areas simply via the aggregation of census sections or the inspection of the municipal registers of inhabitants or other sources of intra-municipality information. Actually, just in the case of a few municipalities was it possible for us to gather statistical information regarding dispersed residential areas. Finally, we had to resource to fieldwork in order to complete our territorial analysis, something which proved also useful in order to put to the test the municipal typology we were using and to select the urban residential complexes where our interviews would subsequently be carried out. The combination of these three strategies produced a methodological approach properly adjusted to the objectives of our research.

New text included is written in red color in the paper.  Highlighted in yelow, it’s possible to find text that has been moved. English language and style have been checked.

Round 2

Reviewer 1 Report

I appreciate the work done by the authors in improving the readability of the manuscript. The overall quality of the paper improved.